# Effects of the Loss of Binocular and Motion Parallax on Static Postural Stability

**DOI:** 10.3390/s23084139

**Published:** 2023-04-20

**Authors:** Keita Ishikawa, Naoya Hasegawa, Ayane Yokoyama, Yusuke Sakaki, Hiromasa Akagi, Ami Kawata, Hiroki Mani, Tadayoshi Asaka

**Affiliations:** 1Graduate School of Health Sciences, Hokkaido University, Sapporo 060-0812, Japan; inikeita090807141@gmail.com (K.I.); ayane21yo@gmail.com (A.Y.); yellow-elephant1005@eis.hokudai.ac.jp (Y.S.); akkan-hiro_8527@eis.hokudai.ac.jp (H.A.); chiroru259@eis.hokudai.ac.jp (A.K.); 2Department of Rehabilitation Sciences, Faculty of Health Sciences, Hokkaido University, Sapporo 060-0812, Japan; ask-chu@hs.hokudai.ac.jp; 3Faculty of Welfare and Health Science, Oita University, Oita 870-1124, Japan; mani-hiroki@oita-u.ac.jp

**Keywords:** binocular parallax, head-mounted display, motion parallax, postural control, static balance, virtual reality, wearable technology

## Abstract

Depth information is important for postural stability and is generated by two visual systems: binocular and motion parallax. The effect of each type of parallax on postural stability remains unclear. We investigated the effects of binocular and motion parallax loss on static postural stability using a virtual reality (VR) system with a head-mounted display (HMD). A total of 24 healthy young adults were asked to stand still on a foam surface fixed on a force plate. They wore an HMD and faced a visual background in the VR system under four visual test conditions: normal vision (Control), absence of motion parallax (Non-MP)/binocular parallax (Non-BP), and absence of both motion and binocular parallax (Non-P). The sway area and velocity in the anteroposterior and mediolateral directions of the center-of-pressure displacements were measured. All postural stability measurements were significantly higher under the Non-MP and Non-P conditions than those under the Control and Non-BP conditions, with no significant differences in the postural stability measurements between the Control and Non-BP conditions. In conclusion, motion parallax has a more prominent effect on static postural stability than binocular parallax, which clarifies the underlying mechanisms of postural instability and informs the development of rehabilitation methods for people with visual impairments.

## 1. Introduction

The sensory input required to maintain postural stability in humans includes visual, vestibular, and proprioceptive information. Vision plays an essential role in postural stability [1,2]. Indeed, body sway increases by 20 to 70% when part of the visual information is blocked during a static stance [3]. In addition, a previous study investigating visual contributions to postural stability reported that depth information is one of the most influential types of visual input [4]. According to a recent survey in Japan, the most common causative disease of visual impairment is glaucoma (28.6%), followed by retinitis pigmentosa (14.0%), diabetic retinopathy (12.8%), and macular degeneration (8.0%) [5]. Glaucoma is a group of progressive optic neuropathies characterized by slowly progressive degeneration of retinal ganglion cells and their axons [6]. This degeneration results in asymmetric visual field loss [7], imposing a negative effect on depth information through impaired binocular parallax [8].

Depth information is typically provided by two visual systems: the binocular parallax and the motion parallax. Binocular parallax is the disparity between an object’s retinal images resulting from the horizontal separation of the eyes [9]. It allows an individual to perceive depth information from these disparities. Previous studies on binocular parallax reported that optimal binocular vision provides more information for maintaining postural balance than monocular vision and results in greater postural stability [3,10]. However, other studies have shown no statistical difference [11] or contradictory effects [12] between binocular and monocular vision. Similar to the binocular parallax, the change in the observation point generated by body or head movements inducing optical motion on the retina, termed motion parallax, also provides depth information [13].

In contrast to the binocular parallax, there is limited information on the effects of motion parallax on postural stability. A previous study showed that the postural system uses motion parallax to reduce body sway during quiet standing; however, this effect was observed only for lateral sway [14]. Therefore, the effects of binocular and motion parallax on postural stability are not fully understood. In addition, it is possible that motion parallax, informed by inertial head movements, compensated for the effects of binocular parallax on postural stability reported in previous studies using a monocular vision condition. In contrast, the monocular vision condition may influence both binocular and motion parallax. Therefore, it is crucial to investigate the effects of one parallax solely on postural stability when removing the other parallax for understanding postural instability in people with visual impairments.

Over the past decade, virtual reality (VR) technology has been used as a tool in research [15,16] and rehabilitation training [17,18]. Head-mounted displays (HMD) can enhance the immersive nature and stereoscopic view of computer-based environments [19]. A previous study showed that VR-based interventions, similar to conventional interventions, affect multiple sensory functions, including vision, vestibular function, and proprioception [20]. Thus, VR technology allows users to induce realistic behaviors through virtual stimulation. Moreover, a VR system with an HMD independently manipulates the images on each lens and synchronizes the visual output with the user’s head movements. In other words, a VR system with an HMD can inhibit the perception of binocular and/or motion parallax.

Therefore, the purpose of this study was to clarify the underlying mechanisms of postural instability in people with visual impairment by investigating the effects of the loss of binocular and motion parallax on static postural stability. The hypotheses were: (1) a loss of depth information will induce postural instability, and (2) binocular parallax will affect postural stability in the anteroposterior (AP) direction because the sensitivity of binocular parallax decreases according to the distance between the object and the individual [21].

## 2. Materials and Methods

### 2.1. Participants

An a priori power analysis was performed using the F-test model of G* Power 3.1 based on a pilot study conducted with five participants (calculated effect size, *f* = 0.33). The necessary sample size for this study was determined to be *n* = 21 to achieve a 0.95 statistical power level. A previous study showed that 20 individuals (27.0%) showed abnormal binocular vision in the cohort of healthy young adults with no history of eye disease or medication (n = 74) [22]. Therefore, the final sample size was determined by adding 30% to account for the likelihood of abnormal stereoacuity, dropouts, and outliers (*n* = 28).

A total of 28 healthy young adults were recruited for the study. The inclusion criteria were as follows: (a) an absence of any neurological or musculoskeletal disorders, (b) an absence of any uncorrected vision problems that affected their ability to follow the testing procedures, and (c) no experience with VR environments. All participants signed informed consent forms approved by the Institutional Review Board of the Faculty of Health Sciences of Hokkaido University (No. 21-20). The study was conducted in accordance with the Declaration of Helsinki (1964).

### 2.2. Equipment

A force plate (Kistler, Winterthur, Switzerland) was used to calculate the center-of-pressure (COP) coordinates in the anteroposterior (AP) and mediolateral (ML) directions. Force plate data were collected at a sampling frequency of 1000 Hz and filtered using a fourth-order 8 Hz low-pass zero-lag Butterworth filter. A head-mounted display (VIVE Pro Eye HMD, weight: 440 g; HTC, New Taipei City, Tai-wan) was used to present the four types of visual conditions with different viewing angles in the virtual reality environment. Unity (Unity Technologies, San Francisco, CA, USA) and Visual Studio (Microsoft, Redmond, WA, USA) were used to create the visual conditions.

The visual background consisted of a 200 × 300 cm picture (67 × 90° of visual angle) of a house with a garden, as described in a previous study [23] (Figure 1). The visual background was placed 1500 cm away from the participant’s eyes. In addition, a window frame (300 × 240 cm) was placed 500 cm from the participant’s eyes between the visual background and the subject to equalize the sensitivity of the binocular parallax to the motion parallax [21]. The four visual test conditions included: normal vision (Control), no-motion parallax (Non-MP), no binocular parallax (Non-BP), and absence of both motion and binocular parallax (Non-P). The same image was projected onto the participants’ left and right eyes during testing to eliminate the motion parallax (Non-MP and Non-P test conditions). In contrast, the binocular parallax was nullified by linking the visual background and the participants’ head movements (Non-BP and Non-P test conditions).

### 2.3. Procedure

All participants were first tested for stereoacuity using the Titmus circles test (Stereo Optical Company, Chicago, IL, USA) to exclude uncorrected vision problems. This stereoacuity test is a commonly used counter test measuring 800–40 s of arc. Participants wearing polarized glasses were asked to identify the circle in each grouping that appeared to “pop up” above the level of the others. The test was stopped when an incorrect response was given, and stereoacuity was recorded as the last level correctly identified. The numerator represents the number of correct answers. Stereoacuity is usually expressed as 1/9 for 800 s of arc and 9/9 for 40 s of arc, meaning that a lower number of seconds of arc indicates better stereoacuity. Since bifoveal fixation is required to achieve a stereoacuity of 60 s of arc [24], we required a test result of 60 s of arc (7/9) or better to include participants in the analysis.

After the stereoacuity test, the participants practiced quiet standing on a high-density (50 kg/m^3^) closed-cell foam pad (47 × 39 × 6 cm; Airex, Sins, Switzerland) for at least 1 min to minimize the effects of spontaneous motor adaptation. The HMD was then attached to the participant’s head, and the participant experienced the four visual conditions previously described while sitting on a chair. After the familiarization period, all participants performed quiet standing tasks under the four visual conditions (Control, Non-BP, Non-MP, and Non-P). For each condition, the participants stood on a foam pad attached to the force plate with their feet together and arms at their sides for 60 s. They were asked to “stand still and relaxed”, and to “gaze at a cross placed at the center of the window frame” (Figure 1).

In each session, five trials were performed under each condition, resulting in a total of twenty trials over four sessions. The order of the visual conditions was randomized for each participant, but the same visual conditions were presented five times in each session. The participants were allowed to rest for at least 5 min between sessions to avoid fatigue and motion sickness.

### 2.4. Data and Statistical Analysis

All signals were processed offline using MATLAB software (MathWorks, Natick, MA, USA). The force plate data were collected for 60 s, with the first and last 5 s removed from the analysis to eliminate the effects of the initiation and end of the visual conditions. The primary outcome was the COP data collected from the force plate. A 95% confidence area of the COP in the horizontal plane (Sway area) was calculated under each visual condition to evaluate the postural stability. The sway area indicates 95% of the variance value when the axis is transformed in the eigenvector direction for each direction, which is expected to enclose approximately 95% of the points of the COP displacements [25]. In addition, the mean velocity and range of the COP displacements in the AP and ML directions (Velocity AP, Velocity ML, Range AP, and Range ML) were calculated to evaluate the directional specificity of postural control. Velocity is the distance per second in each direction, and Range refers to the distance between the positions measured as the furthest distance from the center in each direction [26]. The average of each postural measurement across five trials in each visual condition was calculated and used for the statistical analysis.

A logarithmic transformation was used for all postural stability measurements to confirm normal distribution. A one-way repeated-measure analysis of variance (ANOVA) was used with the results collected under each visual condition (Control, Non-BP, Non-MP, and Non-P) to analyze possible differences in postural stability measurements. Post-hoc analysis was performed using Bonferroni pairwise comparisons. Statistical analysis of the postural stability measurements was performed using SPSS Statistics (version 27.0; IBM, Armonk, NY, USA). Statistical significance was set at *p* < 0.05. For each significant main effect, the effect size was calculated using eta squared (*η*^2^) for the ANOVA models, and Cohen’s d for the follow-up *t*-test among visual conditions.

## 3. Results

Owing to the results of the Titmus circles test, four participants were excluded from all data analyses. The final sample size consisted of 24 healthy young adult participants (14 males and 10 females, age: 22.9 ± 1.2 years; height: 167.9 ± 8.3 cm; weight: 60.2 ± 10.1 kg; a median score of the Titmus circles test: 40). Four participants wore glasses, and 15 wore contact lenses with HMD during the experiments. In addition, none of the participants reported fatigue or adverse events, such as motion sickness, caused by the VR environment.

Figure 2 shows the COP trajectories of a representative trial during a quiet stance under each visual condition. An increased Sway area was observed under Non-MP and Non-P conditions compared with the other two conditions (Control and Non-BP).

A significant main effect of visual conditions was observed for all postural stability measurements (Table 1). Participants showed significantly larger Sway areas under both the Non-MP and Non-P conditions compared to those under the Control (*p* = 0.001 and *p* = 0.003, respectively) and Non-BP conditions (*p* < 0.001 and *p* < 0.001, respectively; Figure 3A and Table 2). Similarly, for direction-specific measurements, the range and mean velocity of COP displacements in each direction were significantly higher under the Non-MP and Non-P conditions than under the Control conditions (Range AP: *p* = 0.030 and *p* = 0.033; Range ML: *p* = 0.003 and *p* = 0.004; Velocity AP: *p* = 0.015 and *p* = 0.004; Velocity ML: *p* = 0.001 and *p* = 0.001, respectively; Figure 3B–E and Table 2). The range and mean velocity of the COP displacements in each direction were also significantly higher in the Non-MP and Non-P conditions than in the Non-BP condition (Range AP: *p* = 0.013 and *p* = 0.021; Range ML: *p* < 0.001 and *p* < 0.001; Velocity AP: *p* = 0.014 and *p* = 0.003; Velocity ML: *p* < 0.001 and *p* < 0.001, respectively; Figure 3B–E, Table 2). However, there were no significant differences between the postural stability measurements under the Control and Non-BP conditions or between those under the Non-MP and Non-P conditions.

## 4. Discussion

This study revealed the effects of binocular and/or motion parallax loss on postural stability during static standing using a VR system with an HMD. The loss of motion parallax or the loss of both binocular and motion parallax induced static postural instability in both the AP and ML directions. However, participants showed no significant changes in the spatial and temporal components of static postural stability when only the binocular parallax was removed. These results indicate that motion parallax can provide more crucial depth information than binocular parallax during static standing.

### 4.1. Contribution of Parallax to Postural Stability

Consistent with a previous study [14], our findings showed a decreased sway area while standing with motion parallax (with or without binocular parallax) compared to that while standing without motion parallax. The sway area (95% confidence area of the COP) during a static stance provides a measure of postural instability [27,28]. Therefore, the result of this study indicated that postural instability was induced by the lack of motion parallax while standing quietly. Motion parallax can be generated even when individuals make unintentional head movements, which allows them to judge the distances to an object and unconsciously stabilize their head movements [13,29]. Therefore, in our experimental conditions, participants received and utilized depth information between themselves and a window or picture through motion parallax to stabilize their posture, even while standing bipedally. Our study results showed no significant differences in the postural stability measurements between the Control and the Non-BP conditions. Specifically, the range of COP is affected by the difficulties of visual sensation rather than vestibular and proprioceptive sensation [25]. In other words, no change in the range of COP between the Control and the Non-BP condition would reflect that visual sensation has little effect on postural stability as the change of these visual conditions. Therefore, these findings suggest that motion parallax can compensate for the lack of binocular parallax to maintain postural stability in a quiet stance. Thus, devices or rehabilitation methods using motion parallax to the fullest extent may improve the postural instabilities of people with visual impairments, especially people with glaucoma, which selectively impair binocular parallax.

In contrast, all postural stability measurements under the Non-MP conditions were higher than those under the Control conditions and showed no significant changes compared to those under the Non-P conditions. One explanation for these results may be that the participants had never experienced a visual condition similar to the Non-MP conditions. Participants under the Non-MP conditions perceived their body and head movements through proprioceptive and vestibular information. but However, they did not perceive motion parallax with full vision, creating a contradiction between their vision and other sensations. The velocity of COP reflects the number of adjustments performed by the postural control system [28], which indicates an ability to modulate/integrate the three sensation systems (visual, vestibular, and proprioceptive) [30]. Thus, the increased velocity under the Non-MP and Non-P conditions could be interpreted as the increased vestibular and proprioceptive system’s weighting to compensate for the visual system with the absence of motion parallax. Previous studies have shown that the central nervous system (CNS) suppresses visual information [31,32,33] and increases the contribution of both the vestibular and proprioceptive systems to maintain postural stability with impaired vision; however, the CNS cannot fully compensate for appropriate visual information [1,34]. Therefore, participants under the Non-MP and Non-P conditions experienced suppressed reliance on visual information, which resulted in decreased postural stability.

In addition, the distance between the participants and objects in the VR environments may explain why the binocular parallax did not compensate for the lack of motion parallax. A previous study investigating the effects of binocular vision during a static stance with near (40 cm) or far (200 cm) distances between the participant and an object reported that postural sway increased at a far distance compared to a near distance with binocular vision but not monocular vision [12]. The authors concluded that the sensitivity of the binocular parallax decreased and caused an impaired function in detecting depth information at far distances. However, no significant changes in postural stability were observed between near and far distances with binocular vision using a convergent prism to equalize eye movements between near and far distances [35]. These findings indicate that proprioceptive information related to eye movements contributes more to maintaining postural stability than the binocular parallax caused by different images on each retina. In fact, several studies have shown that the gazing function affects postural stability more than visual stimulation [36,37]. Therefore, the distance in our experimental conditions (500 cm) may have been too large to induce sufficient eye movements to provide accurate depth information [38]. Notably, many previous studies investigating static postural stability have used a visual target. This finding suggests that researchers should consider the distance between the participant and the visual target. For example, a visual target settles less than 200 cm ahead of the participant to minimize the negative impact of binocular parallax.

### 4.2. Directional Specificity of Postural Instability by a Loss of Parallax

The study results showed that a loss of motion parallax induced postural instability, including an increased Sway area and Velocity in both the AP and ML directions. However, this is inconsistent with a previous study reporting that motion parallax only contributes to postural stability in the ML direction [14]. These different findings may be explained by the use of a firm surface in the previous study and a foam surface in this study. A foam pad was used to reduce the precision of proprioceptive information and increase the reliance on visual information to maintain postural stability [39,40]. Furthermore, previous studies have shown that head movements in the AP direction significantly increase when standing on a foam surface compared with a firm surface, even in healthy young adults [41,42]. Therefore, standing on the foam pad allows healthy young participants to enhance the sensitivity of motion parallax in the AP direction, owing to increased head motion. A previous study showed that vision contributed more to controlling head movement than hip and knee movements (i.e., lower body segments) on a foam surface [41]. However, motion parallax is caused by body or head movements that induce optical motion on the retina in the horizontal plane [43,44]. Thus, depth information obtained through motion parallax could be useful for maintaining postural stability in the ML direction rather than in the AP direction. Our finding supports that the effect sizes of the Range and Velocity between the Non-BP and Non-P conditions in the ML direction were larger than those in the AP direction (Table 2).

However, our findings did not show any significant effects from the loss of binocular parallax in either the AP or ML direction. These findings do not support the hypothesis that binocular parallax affects postural stability in the AP direction. As mentioned in the section “Contribution of parallax to postural stability”, proprioceptive input related to eye movements is crucial compared to the binocular parallax to maintain postural stability under binocular vision. Therefore, the decreased proprioceptive signals related to eye movements due to the distance (500 cm) between our participants and the objects could explain the lack of binocular parallax effects on postural stability in the AP direction. Indeed, increased sway area and velocity with binocular vision were observed even at a distance of 200 cm compared with 40 cm [12]. Moreover, the characteristics of the binocular parallax may explain why there was no significant change caused by a loss of binocular parallax in postural stability in the ML direction. The different images on each retina cause binocular parallax at a particular time. In addition, pupillary distance is constant in everyone. However, head/body movements can create depth information, motion parallax, and perception of one’s body movements over time. Therefore, motion parallax is more effective than binocular parallax for perceiving body movements in the ML direction.

### 4.3. Limitations

This study has several limitations. First, using a foam pad effectively induced reliance on visual sensations; however, it also increased postural instability, which may have been advantageous for motion parallax rather than binocular parallax. Second, motion sickness induced by VR environments would worsen postural stability. However, this potential disadvantage would have little effect on our findings because no participants reported motion sickness while observing the VR environments. Therefore, the potential impact of motion sickness would not influence the comparison in the present study. Third, the HMD used in this study weighed 440 g, which may have added postural instability in both the AP and ML directions. However, a previous study showed that increased postural sway when wearing an HMD was only 2 to 8% [45]. Fourth, our participants first experienced VR environments in this study, which may under- or overestimate the effects of losing depth information in VR environments. Finally, this study was performed with healthy young adults. The effects of the loss of binocular or motion parallax should be investigated in older people and those with visual impairments, especially individuals with glaucoma, given its prevalence in the Japanese population.

## 5. Conclusions

This study showed the effects of an absence of motion or binocular parallax on static postural stability using a VR system, which can control one parallax without influencing the other. The results of this study revealed that a loss of motion parallax induced a significant reduction of postural stability in both the AP and ML directions while standing on a foam surface. In contrast, eliminating the binocular parallax did not impair postural stability. Therefore, our findings suggest that motion parallax can compensate for the loss of binocular parallax, while binocular parallax is not enough to satisfy the absence of motion parallax. In other words, motion parallax may be more important for static postural stability rather than binocular parallax. Moreover, our results suggest that future studies focused on postural stability consider the distance of the visual target to use depth of information effectively. These findings will help develop rehabilitation and assessment methods for people with visual impairments. Further research is needed to investigate the relationship between the impairments of motion parallax solely and postural instability in people with visual impairments and older adults and how to enhance or use motion parallax effectively in people with visual impairments to improve postural instabilities.

## Figures and Tables

**Figure 1 sensors-23-04139-f001:**
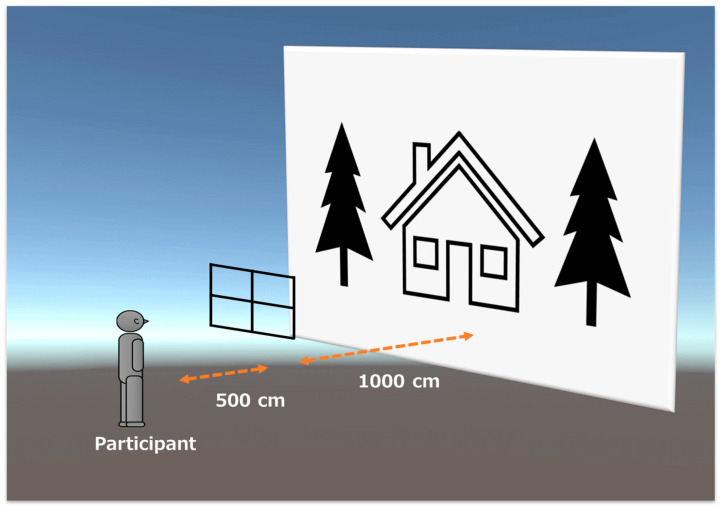
Illustration showing the experimental setup of virtual reality. The visual background consisted of a house with a garden placed 1500 cm away from the participant. The participant was asked to gaze at a cross placed at the center of the window frame set 500 cm away from them.

**Figure 2 sensors-23-04139-f002:**
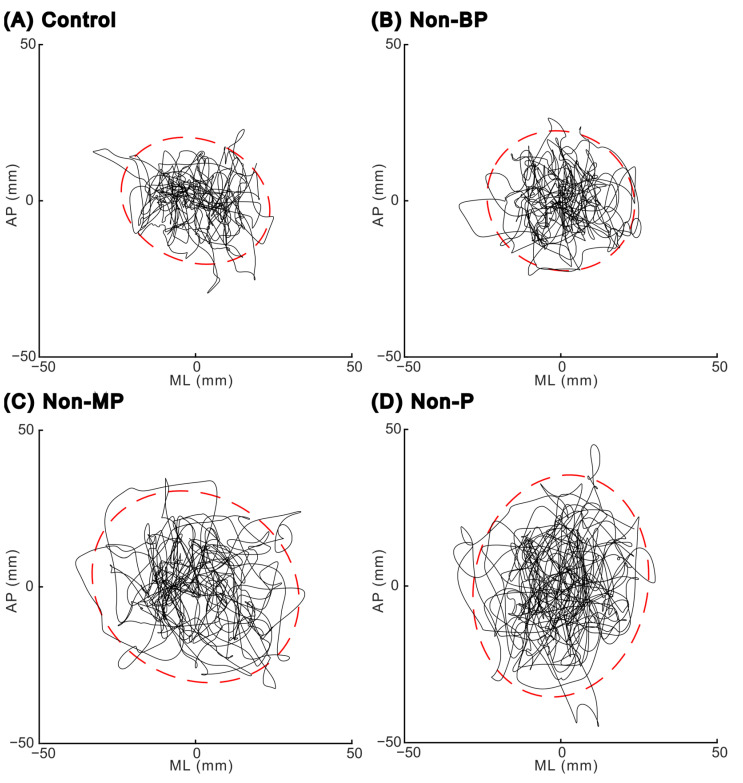
Center-of-pressure trajectories in the horizontal plane for one representative trial in each visual condition: (**A**) Control condition, (**B**) Non-BP condition (without binocular parallax), (**C**) Non-MP condition (without motion parallax), and (**D**) Non-P condition (without both binocular and motion parallax). Red dashed lines indicate the edge of the 95% confidence area of the center of pressure. AP, anteroposterior; ML, mediolateral.

**Figure 3 sensors-23-04139-f003:**
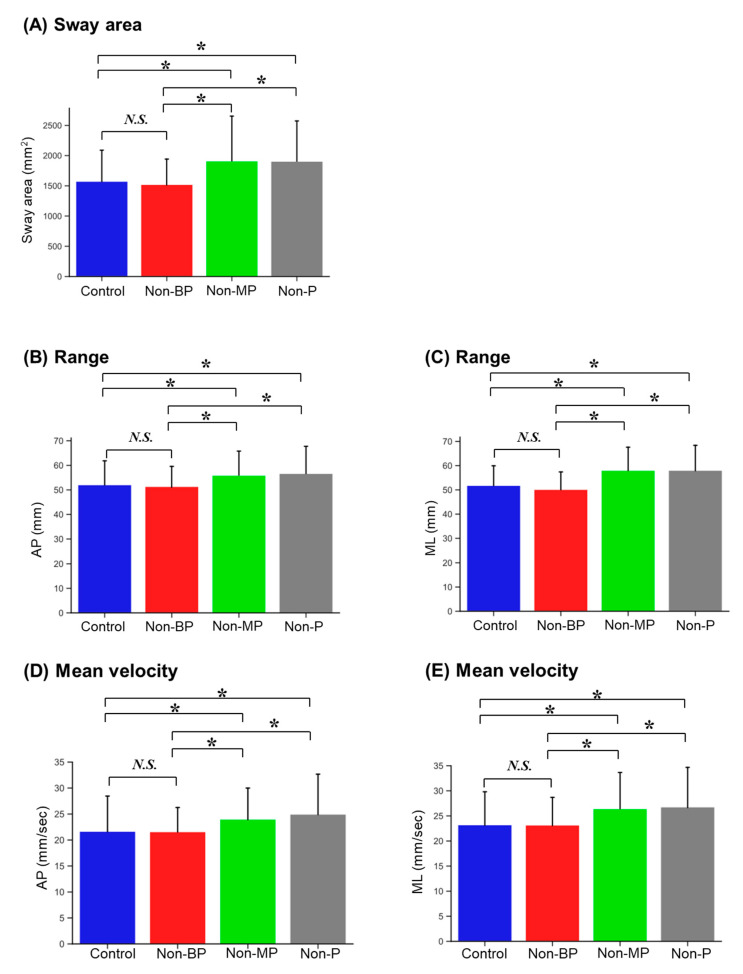
Bar plot of postural stability measurements in each visual condition. Mean values are given: (**A**) sway area, (**B**) range in the anteroposterior (AP) direction, (**C**) range in the mediolateral (ML) direction, (**D**) mean velocity in the AP direction, and (**E**) mean velocity in the ML direction. Error bars indicate standard deviations. *N.S.*, not significance; *, *p* < 0.05.

**Table 1 sensors-23-04139-t001:** Results of postural stability measures using ANOVA.

	Control		Non-BP		Non-MP		Non-P		ANOVA	
	Mean	*SD*	Mean	*SD*	Mean	*SD*	Mean	*SD*	*p* Value	*η* ^2^
Sway area (mm^2^)	1574.2	526.4	1522.1	427.9	1910.6	759.5	1905.3	683.9	**<0.001**	**0.394**
Range AP (mm)	52.1	10.0	51.4	8.4	55.9	10.1	56.7	11.3	**0.001**	**0.220**
Range ML (mm)	51.8	8.3	50.2	7.4	58.1	9.7	58.1	10.5	**<0.001**	**0.422**
Velocity AP (mm/s)	21.7	6.9	21.6	4.8	24.0	6.1	25.0	7.9	**<0.001**	**0.300**
Velocity ML (mm/s)	23.2	6.7	23.2	5.7	26.4	7.4	26.8	8.1	**<0.001**	**0.433**

Visual conditions compared using a one-way repeated-measure analysis of variance (ANOVA) and significant level of 0.05. Bold values indicate significant effects at *p* < 0.05. SD, standard deviation; AP, anteroposterior; ML, mediolateral.

**Table 2 sensors-23-04139-t002:** Results of post-hoc analysis.

	Ct vs. Non-BP		Ct vs. Non-MP		Ct vs. Non-P	
	*p* Value	Cohen’s *d*	*p* Value	Cohen’s *d*	*p* Value	Cohen’s *d*
Sway area (mm^2^)	1.000	−0.175	**0.001**	**0.680**	**0.003**	**0.616**
Range AP (mm)	1.000	−0.095	**0.030**	**0.594**	**0.033**	**0.587**
Range ML (mm)	1.000	−0.262	**0.003**	**0.785**	**0.004**	**0.763**
Velocity AP (mm/s)	1.000	−0.020	**0.015**	**0.576**	**0.004**	**0.762**
Velocity ML (mm/s)	1.000	−0.019	**0.001**	**0.814**	**0.001**	**0.869**
	**Non-BP vs. Non-MP**		**Non-BP vs. Non-P**		**Non-MP vs. Non-P**	
	***p* Value**	**Cohen’s *d***	***p* Value**	**Cohen’s *d***	***p* Value**	**Cohen’s *d***
Sway area (mm^2^)	**<0.001**	**0.827**	**<0.001**	**0.892**	1.000	−0.008
Range AP (mm)	**0.013**	**0.705**	**0.021**	**0.631**	1.000	0.082
Range ML (mm)	**<0.001**	**1.152**	**<0.001**	**1.285**	1.000	<0.001
Velocity AP (mm/s)	**0.014**	**0.635**	**0.003**	**0.725**	1.000	0.215
Velocity ML (mm/s)	**<0.001**	**0.948**	**<0.001**	**1.008**	1.000	0.097

Post-hoc analysis was performed using Bonferroni pairwise comparisons. Bold values indicate significant differences between visual conditions. Ct, control; AP, anteroposterior; ML, mediolateral.

## Data Availability

The data presented in this study are available on request from the corresponding author.

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
