# Peer review of "Effects of the Loss of Binocular and Motion Parallax on Static Postural Stability"

_sensors, 2023, doi:10.3390/s23084139_

Round 1

Reviewer 1 Report

Dear authors,

I feel that your work is well structured. All the doubts I had were addressed in the limitations section.

One minor comment is regarding the implications of your results. Please discuss the implications and practical applications of your findings further in the discussion and not only in one sentence in the conclusion section.

Best regards

Author Response

Dear Reviewers and Editors,

Thank you for carefully considering our manuscript and the beneficial recommendations and feedback. We have modified our manuscript in response to the reviewers’ concerns, and in the attached cover letter, we gave a point-by-point reply to the reviewers’ recommendations. The changes in the revised manuscript are highlighted in red.

On behalf of all authors,
Thank you for a thorough review.

Reviewer 2 Report

The study describes the experimental setup and procedures used to examine the effect of binoculars and motion parallax on postural stability. The sample size of 28 participants is justified based on a priori power analysis, which accounted for the likelihood of abnormal stereo acuity, dropouts, and outliers. The study adhered to ethical guidelines and obtained informed consent from participants.

  1. Regarding the sample size, while the authors performed a power analysis to determine the necessary sample size, the use of a pilot study with only five participants to calculate the effect size may not be representative of the population. Additionally, adding 30% to account for dropouts and outliers seems arbitrary and may not accurately reflect the potential impact on the results.

  2. The study design could benefit from a control group with no visual stimuli, as it would help determine visual cues' contribution to postural control. Without a control group, it is difficult to attribute the effects solely to the absence of motion and binocular parallax.

  3. The authors state that participants with uncorrected vision problems were excluded, but there is no mention of how the vision was assessed or corrected for those who required it. Uncorrected vision problems may affect postural control, and this potential confounding factor should be addressed.

  4. The use of virtual reality as a means of manipulating visual stimuli is an innovative approach. Still, the validity of the virtual environment in replicating real-world conditions should be addressed. Additionally, the authors should consider the potential impact of motion sickness on postural control in virtual reality.

  5. The authors should provide more information about the analysis methods used to evaluate postural control, such as the specific variables measured and statistical tests performed. Additionally, the interpretation of the results should be supported with appropriate references to prior research in the field.

Author Response

(The authors gave the same response as above.)

Reviewer 3 Report

Journal Name: Sensors-MDPI

Title: Effects of the loss of binocular and motion parallax on static postural stability

In the current research, L. Hasegawa et al., has did a detailed research on static postural stability. This work is interesting and quite matches with the scope of MDPI sensors. However, the article needs to be improved further to reach the general audience of MDPI sensors. Based on the above considerations, a major revision is given. Here are some suggestions.

Major revision

1.      The authors have to cite relevant MDPI sensors

2.      How the article is suitable for MDPI sensors has to be explained in a brief way.

3.      Resolution of Fig 2 has to be improved

4.      What was the main research question you were trying to answer in this study, and why is it important to investigate the effects of binocular and motion parallax on static postural stability? Has to be explained in the introduction in a brief way

5.      Can authors discuss methods they have used to measure static postural stability

6.      What were the key findings of your research in terms of the impact of binocular and motion parallax on postural stability? Discuss precisely in the conclusion

7.      Were there any limitations or challenges you encountered during the research process, and how did you address them?

8.      In a few lines discuss (what authors think are the most important directions for future research in this area, and how might your study contribute to further advances in our understanding of binocular and motion parallax and their impact on postural control?)

How did authors control for potential confounding variables in your study design, such as age or prior experience with the experimental task? 

Author Response

(The authors gave the same response as above.)

Round 2

Reviewer 3 Report

The author revised the article and all issues are solved article can be accepted in the present form.